# Identification of Growth-Related Gene *BAMBI* and Analysis of Gene Structure and Function in the Pacific White Shrimp *Litopenaeus vannamei*

**DOI:** 10.3390/ani14071074

**Published:** 2024-04-01

**Authors:** Ruigang Niu, Xiaojun Zhang, Yang Yu, Zhenning Bao, Junqing Yang, Jianbo Yuan, Fuhua Li

**Affiliations:** 1CAS and Shandong Province Key Laboratory of Experimental Marine Biology, Institute of Oceanology, Chinese Academy of Sciences, Qingdao 266071, China; nrg1140458030@163.com (R.N.); yuyang@qdio.ac.cn (Y.Y.); 17864277709@163.com (Z.B.); 15902254601@163.com (J.Y.); yuanjb@qdio.ac.cn (J.Y.); fhli@qdio.ac.cn (F.L.); 2College of Earth Science, University of Chinese Academy of Sciences, Beijing 100049, China; 3Key Laboratory of Breeding Biotechnology and Sustainable Aquaculture, Chinese Academy of Sciences, Wuhan 430072, China

**Keywords:** single nucleotide polymorphism, RNA interference, RNA-seq, lipid metabolism, glucose metabolism

## Abstract

**Simple Summary:**

The growth-related gene *BAMBI* was identified in *Litopenaeus vannamei* through SNP site-specific validation, RNAi, and multi-omics analyses. Its pivotal role in lipid metabolism, glucose metabolism, and protein transport was demonstrated. This study provides a valuable foundation for further exploration and utilization of *BAMBI* genes in shrimp and crustaceans.

**Abstract:**

As one of the most important aquaculture species in the world, the improvement of growth traits of the Pacific white shrimp (*Litopenaeus vannamei*), has always been a primary focus. In this study, we conducted SNP-specific locus analysis and identified a growth-related gene, *BAMBI*, in *L. vannamei*. We analyzed the structure and function of *LvBAMBI* using genomic, transcriptomic, metabolomic, and RNA interference (RNAi) assays. The *LvBAMBI* possessed highly conserved structural domains and widely expressed in various tissues. Knockdown of *LvBAMBI* significantly inhibited the gain of body length and weight of the shrimp, underscoring its role as a growth-promoting factor. Specifically, knockdown of *LvBAMBI* resulted in a significant downregulation of genes involved in lipid metabolism, protein synthesis, catabolism and transport, and immunity. Conversely, genes related to glucose metabolism exhibited significant upregulations. Analysis of differential metabolites (DMs) in metabolomics further revealed that *LvBAMBI* knockdown may primarily affect shrimp growth by regulating biological processes related to lipid and glucose metabolism. These results suggested that *LvBAMBI* plays a crucial role in regulating lipid metabolism, glucose metabolism, and protein transport in shrimp. This study provides valuable insights for future research and utilization of BAMBI genes in shrimp and crustaceans.

## 1. Introduction

The Pacific white shrimp, *Litopenaeus vannamei*, provides more than 80% of the global total shrimp production [1], and plays a crucial role in both the marine ecosystem and mariculture industry. In recent decades, significant efforts have been made to improve the economic traits of *L. vannamei*, including growth rate, disease resistance, and tolerance to extreme temperatures and salinity levels [2,3,4]. Among these traits, the improvement of growth characteristics has been the primary focus of shrimp selection and breeding endeavors.

Growth traits in shrimp, such as body weight and length, exhibit a remarkably high degree of heritability [5,6], with offspring showing genetic gains of up to 10.7% [7], surpassing those observed in farmed terrestrial animals. The advent of high-throughput SNP typing technology in aquatic animals has facilitated the localization and discovery of numerous quantitative trait loci (QTLs) and genes associated with economically important traits. In shrimp, dozens of growth trait candidate genes, including *AMY2, CTSL, PKC-delta, Rap-2a,* have been identified through QTL mapping and GWAS screening [8,9].

Growth traits are the result of a complex network of genes regulated by multiple factors, however, research on growth genes and their regulation in crustaceans remains limited. In insects, growth and development is mainly regulated by signaling pathways such as insulin, mTOR, TGF-β, MAPK, Ras, Wnt, ecdysone, and others involving hundreds of genes [10,11,12,13]. These well-established growth-related pathways represent an important and viable approach to explore the regulatory mechanisms governing crustacean growth. The structural and functional analysis of *IGFBP* in shrimp confirms its significant impact on shrimp growth [14]. In addition, the mTOR pathway gene, *Raptor*, may actively participate in transmitting nutritional stimulation signals to downstream effector S6K1, thereby maintaining cell size and mTOR protein expression, ultimately affecting growth in *L. vannamei* [15].

The TGF-β signaling pathway is involved in various cellular processes such as cell growth, differentiation, apoptosis, motility and invasion, and extracellular matrix (ECM) production. Key components of this pathway include TGF-β ligands, TGF-β receptor, Smad proteins, and BMP molecules. This complex cellular signaling cascade undergoes precise regulation at multiple levels including ligand-receptor interactions, Smad-mediated events, and transcriptional control within the nucleus [16,17,18]. The TGF-β signaling pathway is a closely related pathway to crustacean growth. For instance, in the Chinese shrimp *Fenneropenaeus chinensis*, the inhibitory protein Myostain (Mstn), a member of the TGF-β family protein, has been found to exert a crucial regulatory influence on shrimp growth [19]. Knockdown of *FcMstn* has significant effects on TGF-β ligands, TGF-β receptors and key genes such as *Smad*s and *Ras* [20]. However, investigations regarding the TGF-β pathway in crustaceans remain limited. 

BAMBI, also known as BMP and activin membrane-bound inhibitor, shares structural similarity with the TGF-β type I receptor, however, it lacks the intracellular Ser/Thr kinase structural domain. Therefore, it is commonly regarded as a pseudoreceptor within the TGF-β pathway. BAMBI forms stable complexes with TGF-β type II receptors, and effectively inhibits BMP and activin signaling along with TGF-β signaling [21,22]. Moreover, BAMBI exerts an inhibitory effect on TGF-β while enhancing Wnt/β-catenin signaling in diverse cell types [23], thereby serving as a critical regulator of cell proliferation and differentiation. In vertebrates, *BAMBI* is highly expressed in tissues such as the adrenal gland, ovary, and placenta (https://www.ncbi.nlm.nih.gov/gene/25805, accessed on 14 October 2022), and most of the studies on the *BAMBI* gene have focused on the cell proliferation and differentiation, adipose tissue formation, and cellular muscle-forming differentiation. *BAMBI* knockdown promoted the differentiation of bovine preadipocytes and suppressed myoblast myogenesis, as indicated by the increased lipid droplets and the decreased myotubes, as well as the corresponding significant changes in the expression of *PPARγ*, *C/EBPα*, *C/EBPβ*, *FABP4*, *MyoD*, *MyoG* and *Myf6* [24]. Nevertheless, the structure and function of BAMBI in invertebrates is unclear.

In this study, we conducted SNP variation analysis and validation to screen and identify the growth-related gene *BAMBI* in *L. vannamei* individuals with different growth rates. Subsequently, we cloned the gene and analyzed its gene structure, phylogeny and expression pattern. Using RNA interference (RNAi), we successfully knocked down the expression of the *BAMBI* gene and observed changes in growth phenotypes as well as the expression of related genes. Our results suggest that this gene may regulate shrimp growth by modulating various pathways such as lipid metabolism and glucose metabolism. This study provides a preliminary exploration of the potential molecular mechanism underlying the regulation of shrimp growth by the *BAMBI* gene, and provides valuable genetic information for molecular design breeding strategies in shrimp aquaculture.

## 2. Materials and Methods

### 2.1. Experimental Animal

Experimental shrimp were cultured in seawater at the shrimp genetic laboratory of the Institute of Oceanography, Chinese Academy of Sciences (Qingdao, China). The cultured seawater was filtered, sterilized, and continuously oxygenated at a temperature of 25 ± 1 °C, salinity of 3%, and pH of 7.5 ± 0.1. All shrimp were fed the same commercial food pellets (Dale Feed Company, Yantai, China) three times a day at 10 a.m., 4 p.m. and 10 p.m. 

### 2.2. Screening for Growth Trait-Related Genes

Based on the insect growth pathway analyses, we screened a total of 428 genes in the *L. vannamei* genome and transcriptome data that were involved in several growth signaling pathways, such as the insulin pathway, ecdysone pathway, mTOR pathway, TGF-β pathway, and others. The genes were compared in the available SNP dataset of eight resequenced families and their different mutation frequencies were counted.

The resequencing data used in this study were whole genome sequences of 220 individuals of *L. vannamei* obtained in our previous research [25]. All sequences were generated on the Illumina Hiseq2500 and DNBSEQ-T7 sequencing platform [26], and clean reads from each sample were mapped to the *L. vannamei* reference genome using the Burrows-Wheeler Aligner [27]. SNP calling was performed using the GATK’s Variant Filtration with standards (-filter “QD < 2.0 || FS > 60.0 || MQ < 40.0,” -G_filter “GQ < 20”). The high quality SNPs were annotated using ANNOVAR [28]. The VCF file was used for further allele frequency analysis between different samples. Finally, a whole genome resequencing SNP dataset for *L. vannamei* was obtained.

Further screening was performed in the *L. vannamei* resequencing SNP dataset. SNP loci for genes with allele frequencies between 0.05 and 0.95 were selected, and SNP loci with mutation sites located on the same chromosome and with allele frequencies close to 0.95 were prioritised, focusing on meaningful mutations such as non-synonymous mutations and code-shifting mutations.

A total of 25 genes obtained from the above screening were compared with Blastp on NCBI (https://blast.ncbi.nlm.nih.gov/, accessed on 14 October 2022) and then with SMART (https://smart.embl.de/, accessed on 14 October 2022) to predict their structural domains, and it was determined that their function may be related to *L. vannamei* growth.

Primers were designed based on SNP loci to be validated by PCR amplification in a different growth rate population, *L. vannamei* Family 21253. The genotype and mutation frequencies of the 25 target genes’ SNP loci in individuals with different growth rates were counted, and the Chi-Squared Test was performed to test whether there were significant differences in allele frequencies among individuals with different growth rates. If there is a significant difference in the SNP loci, it indicates a close association between the SNP loci and the growth trait of body weight, making them potential molecular markers for growth-related genes. In addition, the differential expression of the screened growth-related genes were analyzed in the transcriptome data (accession number: PRJNA987844) of the populations with different growth rates, and the data were then validated by differential expression analysis using DESeq2 [29].

### 2.3. Analysis of Gene Structure

To analyze the gene structure of *BAMBI* of *L. vannamei*, we extracted the sequences of *BAMBI* from the shrimp genome and the transcriptome data obtained in our previous study. Multiple sequence alignment was performed to identify the complete coding sequence (CDS) of *BAMBI* using Sequencher (https://www.genecodes.com/content/sequencher-546-released-0, accessed on 2 February 2023) and DNAMAN (www.onlinedown.net, accessed on 2 February 2023). The sequence was submitted to the ORF Finder (https://www.ncbi.nlm.nih.gov/orffinder/, accessed on 4 February 2023) and the ExPASy translation tool (https://web.expasy.org/translate/, accessed on 4 February 2023) to deduce its amino acid sequence. To determine the conserved structural domains of *BAMBI*, its amino acid sequence was analyzed using SMART (https://smart.embl.de/, accessed on 5 February 2023). Its protein 3D structure was predicted using the SWISS-MODEL website (https://swissmodel.expasy.org/interactive, accessed on 13 February 2023), and the final results were evaluated for quality using savesv6.0 (https://saves.mbi.ucla.edu/, accessed on 13 February 2023). Motif prediction was performed using the MEME online website (https://meme-suite.org/meme/doc/meme.html, accessed on 21 February 2023). The signal peptide was predicted using the SignalP (https://services.healthtech.dtu.dk/services/SignalP-5.0/, accessed on 23 February 2023). The gene was finally named as *LvBAMBI*. MEGA 7 software (Institute of Molecular Evolutionary Genetics, PA, USA) was utilized for constructing the BAMBI phylogenetic tree using the NJ neighbour-joining method with 1000 bootstrap value tests. The BAMBI phylogenetic tree was beautified using the online website Tree of Life (iTOL) (https://itol.embl.de/, accessed on 2 March 2023) and Photoshop (https://helpx.adobe.com/cn/photoshop/get-started.html, accessed on 3 March 2023).

### 2.4. Prediction of Transcription Factors

The sequence of the first 2000 bp of the *LvBAMBI* transcription start site was extracted to predict the transcription start site (TFBS) using the animalfdb 3.0 program (http://bioinfo.life.hust.edu.cn/AnimalTFDB/#!/, accessed on 14 April 2023). A *p*-value of < 10^–7^ was used as a restriction for the selection of TFBSs. Results were visualized using the website (https://www.omicshare.com/tools/home/report/report_sankey.html, accessed on 14 April 2023).

### 2.5. Gene Expression Pattern Analysis

In our previous studies, we conducted RNA-Seq of 20 early developmental stages and eight molting stages, with the registration numbers SRR1460493-SRR1460505 and SRX1098368-SRX1098375, respectively. Additionally, RNA-Seq was performed on 16 types of adult tissues in *L. vannamei* [25]. The FPKM values of *LvBAMBI* were calculated using the RSEM program (version 1.3.3) by mapping clean reads to unigenes from the transcriptome data. These values were then utilized for comparing gene expression differences. Genes with *p* ≤ 0.01 and an absolute value of log2 (fold change) ≥ 1 were identified as differentially expressed genes (DEGs). Column charts were generated using GraphPad software (GraphPad Software Inc, La Jolla, CA, USA).

### 2.6. In vivo RNA Interference

The BAMBI-GR-F/R primer pair was designed using Primer3Plus. The T7 promoter sequence was added to form dsBAMBI-F and dsBAMBI-R. The dsBAMBI-F/R primer pair was used to amplify a 435 bp full-length cDNA fragment of *LvBAMBI*. The PCR involved annealing at 95 °C for 4 min, followed by 35 cycles of amplification (94 °C for 30 seconds, 59 °C for 30 s, 72 °C for 30 s), and extension at 72°C for 10 min. The same procedure was used to amplify a 289-bp EGFP cDNA fragment, with an annealing temperature of 60 °C. The PCR products were purified using a MiniBEST DNA fragment purification kit (Takara, Japan). The dsRNA was synthesized using the purified product and the Transcript Aid T7 High Yield kit (Thermo Fisher Scientific, Waltham, MA, USA) according to the manufacturer′s instructions. The quality and concentration of dsRNAs were determined using 1.5% agarose gel electrophoresis and NanoDrop 2000 (Thermo Fisher Scientific, Waltham, MA, USA), and qualified dsRNAs were stored at −80 °C. 

To determine the optimal dose of dsRNA interference, 84 subadult *L. vannamei* at D1-D2 pre-molt stages were selected for preliminary experiments. The individuals were divided into seven groups, with each group containing 12 shrimp. There were three experimental groups (dsBAMBI group), three control groups (dsEGFP group), and one PBS group. Each group had three replicate samples, with four individuals in each sample. Different dosages of RNAi (1 µg, 4 µg, and 8 µg) were applied. The dsBAMBI and dsEGFP groups were injected with 1 µg, 4 µg, and 8 µg dsRNA dissolved in 10 µl PBS. The PBS group was injected with only 10 µl PBS. After 48 hours, the RNAi efficiency of each pre-experiment group was determined by collecting the lymphoid organ (Oka), gill, eye-stalk, and hepatopancreas for RNA extraction, cDNA synthesis, and qPCR experiments. The interference efficiency was compared, and a final interference dose of 1.6 µg per g of shrimp injection was determined.

In a formal RNAi experiment, 244 individuals at the D1-D2 stage of pre-molt were divided into three groups: the dsBAMBI group, and the control groups (EGFP and PBS group). The individuals were injected with 7.84 µg dsBAMBI, 7.84 µg dsEGFP, and 10 µl PBS, respectively. Each group had three biological replicates, and injections were repeated every four d for a total duration of 16 d. Before the initial injection, the weight and body length of each shrimp were measured using an analytical balance and a ruler with an accuracy of 0.1 cm. At the end of the experiment, the weight and body length were measured again using the same method. Samples of the Oka, gill, eye-stalk, and hepatopancreas were collected, frozen in liquid nitrogen, and stored at −80 °C.

### 2.7. Real-Time Quantitative PCR

SYBR Green-based quantitative real-time PCR (qPCR) was used to measure the relative expression of *LvBAMBI* in both the experimental and control groups. The internal reference was 18S rRNA. Primer pairs BAMBI-DL-F/R and rt18S-F/R were designed using Pri-mer3Plus (Appendix A). The qPCR was conducted using Eppendorf Mastercycler ep realplex (Eppendorf, Germany). The experiment used the SuperReal PreMix Plus (SYBR Green) kit (Tiangen, China). Each sample had four technical replicates. The qPCR steps were: 94 °C for 2 min, 40 cycles of 94 °C for 20 s, and 62 or 55 °C for 20 s (annealing temperatures of 57 °C and 55 °C for BAMBI-DL-F/R and rt18S-F/R, respectively), and 72 °C for 20 s. The relative expression of *LvBAMBI* was calculated using the 2^−ΔΔCt^ method [30].

### 2.8. RNA-seq Analysis

To analyze the changes in gene expression induced by *LvBAMBI* knockdown, we compared hepatopancreas transcriptome from the RNAi experiment. Nine hepatopancreas samples from each of the dsBAMBI and dsEGFP groups were further divided into three separate groups, serving as three biological replicates for transcriptome sequencing.

Total RNA was used for RNA sample preparation. Briefly, mRNA was purified from total RNA using poly-T oligo-attached magnetic beads (NEBNext® Ultra™ II RNA Library Prep Kit for Illumina®, NEB, Ipswich, MA. USA). First strand cDNA was synthesized using random hexamer primers and M-MuLV Reverse Transcriptase (RNase H-) (NEBNext® Ultra™ II RNA Library Prep Kit for Illumina®, NEB, Ipswich, MA. USA). Second strand cDNA synthesis was then performed using DNA Polymerase I and RNase H (NEBNext® Ultra™ II RNA Library Prep Kit for Illumina®, NEB, Ipswich, MA, USA). Remaining overhangs were converted to blunt ends via exonuclease/polymerase activities, and then fragmented into randomly short fragments with fragmentation buffer. After adenylating of the 3’ ends of the DNA fragments, adaptor with a hairpin loop structure were ligated to prepare for hybridization. In order to preferentially select cDNA fragments 370~420 bp length, the library fragments were purified using the AMPure XP system (Beckman Coulter, Beverly, MA, USA). PCR was then performed using Phusion High-Fidelity DNA Polymerase, Universal PCR primers and Index (X) primers. Finally, the PCR products were purified on the AMPure XP system (Beckman Coulter, Beverly, MA, USA) and the quality of the libraries was assessed on the Agilent Bioanalyzer 2100 system (Agilent Technologies, Santa Clara, CA, USA). After clustering, the library preparations were sequenced on an Illumina Novaseq platform (Illumina, San Diego, CA, USA) at Novogene Corporation Inc (Beijing, China) and 150 bp paired-end reads were generated.

The high-quality clean reads were obtained by removing adapter sequences, reads containing ploy-N and low-quality reads from the raw data. At the same time, the Q20, Q30 and GC content of the clean data were calculated. Finally, the clean reads were compared to the reference genome for sample evaluation using HISAT2 v2.0.5 (https://ccb.jhu.edu/software/hisat/index.shtml, accessed on 1 November 2023).

To quantify expression abundance, a fragment per kilobase of transcript per million mapped reads (FPKM) value was calculated for each transcription region using featureCounts (1.5.0-p3) (https://subread.sourceforge.net/, accessed on 1 November 2023). Differential expression analysis between the experimental and control groups was performed using DESeq2 software (1.20.0) (https://bioconductor.org/packages/release/bioc/html/DESeq2.html, accessed on 2 November 2023). The *p*-value values were adjusted according to the Benjamini & Hochberg method [31]. The corrected *p*-value and |log2foldchange| were used as thresholds for significant differential expression.

The DEGs with *p*-value < 0.05 were analyzed by GO and KEGG enrichment analysis using clusterProfiler (3.8.1) software (https://bioconductor.org/packages/release/bioc/html/clusterProfiler.html, accessed on 2 November 2023) to obtain the biological functions and enriched pathways. The expression of DEGs was visualized using TBtools (V1.098). 

To verify the transcriptome data, ten DEGs were randomly selected and their expression levels were detected by qRT-PCR as described in Section 2.7.

### 2.9. Combined Metabolomic and Lipidomic Analyses

To estimate the changes in metabolic levels induced by *LvBAMBI* knockdown, we selected nine hepatopancreas samples from each of the dsBAMBI group and the dsEGFP group for metabolome and lipidome analysis.

Metabolome analysis is based on liquid chromatography-mass spectrometry (LC-MS) technology [32,33] for non-targeted metabolomics research, and the experimental process mainly includes metabolite extraction from the samples, LC-MS/MS detection, and data analysis. The first three QCs were used to monitor the instrument status and balance the chromatography-mass spectrometry system before sample injection, and the subsequent three QCs were used to perform segmental scanning, which together with the secondary spectra obtained from the experimental samples were used for metabolite characterization. The QC inserted in the middle of the sample detection was used to evaluate the stability of the system throughout the experiment and to analyze the data for quality control.

The raw files (.raw) obtained from mass spectrometry were first imported into Compound Discoverer 3.3 (CD3.3) software (Thermo Fisher Scientific, Waltham, MA, USA) for spectral processing and database searching to obtain the qualitative and quantitative results of the metabolites, and then the data were subjected to quality control to ensure the accuracy and reliability of the data results. Multivariate statistical analyses of the metabolites, including Principal Component Analysis (PCA) and Partial Least Squares Discriminant Analysis (PLS-DA), were then performed to reveal the differences in metabolic patterns between the different groups. Hierarchical clustering (HCA) and metabolite correlation analysis were used to reveal the relationships between samples and between metabolites and metabolites. Finally, the biological significance of metabolite correlation was explained by functional analysis such as metabolic pathway.

Lipidomic sequencing, like metabolomics, is based on LC-MS technology for lipidomic analysis, and the experimental process mainly includes sample collection, lipid extraction, LC-MS/MS detection, and data analysis.

Different from metabolomics data processing, lipidomics first imports the raw files (.raw) obtained from mass spectrometry into Lipidsearch software (Thermo Fisher Scientific, Waltham, MA, USA), and then performs spectral processing and database searching to obtain the qualitative and quantitative results of lipid compounds, and then performs quality control on the data to ensure the accuracy and reliability of the data results. Next, multivariate statistical analysis, HCA and lipid correlation analysis were performed to reveal the relationships among samples and between lipids. Finally, the biological significance of the lipid compound correlation was explained by functional analysis.

### 2.10. Statistical Analysis

Significant differences in gene expression between experimental and control groups were analyzed using SPSS 22.0 software (https://www.ibm.com/cn-zh/analytics/spss-statistics-software, accessed on 15 November 2023), and significant differences between two treatments were indicated by *p* ≤ 0.05 (*) and *p* ≤ 0.01 (**), respectively.

## 3. Results

### 3.1. Identification of LvBAMBI

Based on the resequencing data of 220 individuals and utilizing the genome and transcriptome data of *L. vannamei*, we identified a total of 74 genes potentially associated with growth that exhibited a high frequency of SNP variation. By searching for the 74 genes in the relevant literature, we screened out 25 genes which were used in the genotypic validation experiments. These genes mainly belong to the ecdysone, insulin, Hippo, Ras, and TGF-β pathways (Appendix A). Subsequent PCR amplification targeting SNP sites within these screened genes was validated in Family 21253 of *L. vannamei*, which showed significant growth trait variation among individuals. Notably, significant differences (*p*-value < 0.05) were observed at several SNP sites within the *BAMBI* gene, confirming that the *LvBAMBI* may be closely linked to growth (Appendix A).

Transcriptome analysis of the *LvBAMBI* gene in populations with different growth rates revealed significant disparities in muscle expression levels. In particular, fast-growing individuals showed a 1.91-fold higher expression level of *LvBAMBI* gene compared to the slow-growing individuals (Appendix A).

### 3.2. Structural Characterization of LvBAMBI

The LvBAMBI gene (XM_027371908.1) underwent clonal sequencing and analysis, revealing an open reading frame (ORF) of 1077 nt in length that encodes a protein consisting of 358 aa with a molecular weight of 39.28 kDa and an isoelectric point (pI) of 7.9(Supplementary Data 1 and 2). Notably, the deduced LvBAMBI protein possesses a highly conserved BAMBI structural domain and a transmembrane structural domain (Figure 1B), which is observed in BAMBI proteins across various species, indicating significant homology between invertebrates and vertebrates.

Multiple sequence alignments of conserved elements between LvBAMBI and BAMBIs from other species revealed a high degree of homology (Figure 1C). The alignment results indicated the presence of eight highly conserved sites spanning amino acids 70aa-119aa, including six cysteine residues (C), one glycine residue (G), and one serine residue (S). The predicted position of Pfam_BAMBI within the BAMBI domain was found to be located between 62aa–149aa (Figure 1B,D). These results suggest that the amino acid residues at these conserved sites may play a crucial role in shaping the functionality of BAMBI. 

The 3D structure prediction analysis of LvBAMBI is shown in Figure 1C, where the white region in Figure 1C(a) represented a component of the predicted 3D structure of the LvBAMBI protein, which had a similar structural pattern to the rectangular region in Figure 1C(b). This structural arrangement potentially corresponds to the binding site of TGF-β type II receptor, as suggested by a previous report [21].

### 3.3. Phylogenetic Analysis of BAMBI Genes

Twenty-six BAMBI protein sequences from different species were retrieved from the NCBI database to construct a phylogenetic tree, in conjunction with the LvBAMBI sequence (Figure 2; Appendix A). The phylogenetic analysis showed that LvBAMBI clustered with BAMBI derived from the snow crab Chionoecetes opilio. This crustacean branch formed a clade with BAMBIs from Insecta and Arachnida, followed by Chondrichthyes, Actinopterygii, Amphibian, and Mammalia. From an evolutionary perspective BAMBI first emerged in the marine bryozoan, Bugula neritina. By comparing the protein structural domains and signal peptide of BAMBI in different species, it was found that the BAMBI gene is highly conserved in most species. It consists of an N-terminal signal peptide (SP(Sec/SPI)), a Pfam_BAMBI protein structural domain, and a C-terminal transmembrane structural domain (Figure 1B). These results suggest that BAMBI functions as a transmembrane protein with an associated signal peptide.

BAMBI exhibits a high degree of conservation, as evidenced by the presence of the N-terminal signal peptides, the Pfam_BAMBI domain and the transmembrane domain across invertebrates to vertebrates. This suggests that the BAMBI gene may possess similar functions in different species. Furthermore, in invertebrates such as bryozoans, insects, crustaceans and echinoderms, the number of motifs within BAMBI genes is relatively small. In vertebrates, there is an increase in the number of motifs. In general, the motif structure of BAMBI genes tends to be complex, and the main Pfam_BAMBI domain and the transmembrane domain are quite conserved.

### 3.4. Transcription Factor Prediction of LvBAMBI

Under more stringent conditions (*p*-value < 10^–7^), a total of 39 transcription factor binding sites (TFBSs) were predicted within the 2000 bp upstream region of the transcription start site of *LvBAMBI* (Figure 3).

The obtained transcription factors (TFs) were categorized into distinct groups. The first group comprises CREB1, MYC, ESR1, EZH2, ERG, AR, etc., which are associated with cell differentiation, proliferation, division, as well as limb muscle formation and adult muscle differentiation. These factors play a crucial role in the growth and development of organisms. The second group is primarily involved in stress response, particularly adaptive cellular responses to hypoxia, with HIF1A being the key factor within this category. The third group is closely related to immune response, and mainly includes KDM5B, IRF3, IRF4, etc. These factors regulate various aspects of the immune system, such as B-cell formation, immune evasion, and the body′s response to injury and inflammation. Lastly, the fourth group focuses mainly on transcriptional regulation, including BRD4 and SP2, etc. (Figure 3).

### 3.5. Expression Pattern of LvBAMBI

During various early developmental stages of *L. vanname*, *LvBAMBI* is initially observed in fertilized eggs, indicating its maternal expression. From the 2-cell stage (C2) to blastula stage (blast), there is a gradual increase in *LvBAMBI* expression until it reaches its peak. Subsequently, after the gastrula stage (gast), there is a significant decrease in gene expression and finally reaches a stable state (Appendix A). Throughout the molting process, *LvBAMBI* exhibits minimal expression at pre-molt stages (D1 and D3), but demonstrates peak expression during post-molt stages 2 (P2) (Appendix A). Regarding tissue distribution analysis, we examined the transcriptional profile of *LvBAMBI* across 16 different adult tissues, which revealed its ubiquitous and low expression characteristics. Notably, Oka, gill and intestine exhibited relatively higher levels of *LvBAMBI* expression (Appendix A).

To validate the precision of the RNA-Seq data from different tissues, we conducted qPCR analysis to determine the relative expression level of *LvBAMBI* in 12 adult tissues of *L. vanname*. The results showed the elevated expression levels of this gene in ovary (Ov), testis (Te), and gill (Gi), while displaying lower expression levels in other tissues (Appendix A). Although these results differ from RNA-seq, if we exclude the gonadal expression data, there is a consistent trend between the expression levels of *LvBAMBI* observed through both RNA-seq and qPCR. This suggests that the variation in gonadal expression may be due to different stages of gonadal development within our sample groups.

### 3.6. RNA Interference of LvBAMBI

In the RNA interference (RNAi) pre-experiment, the dsBAMBI group showed a significant decrease in the relative expression level of *LvBAMBI* compared to the control group (Figure 4A). The optimal interference dose of 4µg resulted in an impressive interference efficiency of 75.5%. The formal RNAi experiment lasted for 16 d, and it was observed that dsBAMBI significantly inhibited the relative expression level of *LvBAMBI* in Oka tissues by 86.9% (Figure 4B). Furthermore, after 16 d of RNAi, there were noticeable effects on the growth characteristics of *L. vannamei*. Specifically, compared to two control groups (PBS group and dsEGFP group), the dsBAMBI group showed significantly lower body length (*p* < 0.05) and body weight (*p* < 0.01) (Figure 4C,D). In addition, the number of molts was higher in the experimental group than in the two control groups. However, no significant mortality was observed in the experimental group. These results show that knockdown of *LvBAMBI* affects the growth and molting of shrimp, but it does not result in shrimp mortality (Appendix A). 

### 3.7. Gene Expression Changes after LvBAMBI Knockdown

To evaluate the changes in gene expression after *LvBAMBI* knockdown, we conducted a comparative transcriptome analysis of the hepatopancreatic tissues between dsBAMBI group and the control group (injected with dsEGFP). The results showed that a total of 269.33 M raw reads were obtained from six sequencing libraries, with an effective data volume of each library ranging between 6.18–7.19 Gb and Q30 for 93.58%–95.36% (Appendix A). All raw reads were deposited at the NCBI Sequential Read Archive (SRA) (PRJNA1062129). The obtained clean reads were aligned to the *L. vannamei* genome database, where more than 86.11% of the clean reads could be successfully mapped to the reference genome (Appendix A).

The expression levels of each gene were quantified based on the RNA-seq data. We identified a total of 420 downregulated and 357 upregulated differentially expressed genes (DEGs) (Figure 5A). To gain further insight into the expression patterns of these DEGs, we conducted cluster analysis. The transcriptome analysis revealed significant differences in gene expression between the dsBAMBI group and the control group (Figure 5B). By applying specific selection criteria (Log2(foldchange) > 1, *p*adj < 0.01), we selected 39 of the most significant DEGs from this set and initially categorized them into nine major functional groups based on their putative functions (Table 1). Compared to the control group, DEGs in the dsBAMBI group were mainly associated with intracellular signal transduction and transportation, carbohydrate metabolism, lipid metabolism, protein metabolism and immune-related processes.

Among them, we observed significant changes in relevant genes involved in lipid metabolism in the dsBAMBI group. For instance, *pancreatic lipase-related protein 2-like* exhibited a significant upregulation, while *sphingomyelin phosphodiesterase-like* and *apolipoprotein D-like* showed significant downregulation. Additionally, several genes associated with immune enhancement and stress response were found to be upregulated in the dsBAMBI group, including *leukocyte elastase inhibitor, beta-1,3-glucan-binding protein, cathepsin L1-like*, and so on. Meanwhile, we also identified a number of genes involved in signal transduction, protein digestion, uptake and translocation that showed significant up- or down-regulation after *LvBAMBI* knockdown, including *metabotropic glutamate receptor 3-like, sorting nexin-18-like*, and *SLC19A1*. Furthermore, our analysis revealed additional significantly down-regulated genes in the dsBAMBI group associated with cellular homeostasis (e.g. *HSP70*, *HSP83*, and *HSP90*) and glucose metabolism (e.g. *hydroxyacid oxoacid transhydrogenase* and *delta-1-pyrroline-5-carboxylate synthase-like*) (Table 1).

We conducted a comprehensive analysis of the functional annotations of these DEGs using the Gene Ontology (GO) database. The results showed that the DEGs were significantly enriched in various biological processes, molecular functions, and cellular compositions. Specifically, we identified 357 upregulated DEGs that were significantly clustered within 301 GO entries, while 420 downregulated DEGs were clustered within 306 GO entries (Appendix A). To provide a concise overview, we extracted and presented the top 30 GO entries for both upregulated and downregulated DEGs (Figure 6A,B). The top 30 GO items of upregulated DEGs mainly included transporter activity, lipid transporter activity, regulation of macromolecular metabolic processes, myosin complex, cytoskeleton, etc. Conversely, the top 30 GO terms of downregulated DEGs included those associated with oxidation-reduction process, sulfur compound metabolic process, protein folding, endoplasmic reticulum, endomembrane system, oxidoreductase activity, ATPase activity, etc.

The pathways enriched by DEGs were analyzed using the KEGG database. We selected the top 20 most significant KEGG pathways to represent up- and downregulated pathways (ListHits>3, *p-*value minimum), respectively (Figure 6C,D). The highly enriched pathways included lysosomal pathway, lipid metabolism pathway, endoplasmic reticulum protein processing, protein transport, and some pathways related to glucose metabolism processes (Appendix A). Among them, the upregulated DEGs significantly enriched circadian rhythm-fly, tyrosine metabolism, lysosome, FoxO signaling pathway, and some glycolysis and glucose metabolism related pathways such as TCA cycle. Downregulated DEGs are mainly enriched in the protein processing in endoplasmic reticulum, sphingolipid metabolism, protein export, lysosome and fatty acid metabolism.

Based on clustering and enrichment analyses, we observed significant impacts on metabolic processes after RNAi of the *LvBAMBI* gene. We conducted a comprehensive investigation of genes involved in lipid metabolism, glucose metabolism, and protein metabolism pathways. Our results showed a noticeable tendency towards downregulation of the expression of genes encoding glucocerebrosidase (GBA), sphingomyelin phosphodiesterase-1 (SMPD1), and ceramide synthetase (CerS) within the lysosomal, sphingolipid metabolism, and endoplasmic reticulum protein processing pathways associated with lipid metabolism (Appendix A). We focused on the DEGs of four pathways associated with lipid metabolism: fatty acid metabolism, fatty acid biosynthesis, fatty acid degradation and fatty acid elongation. We observed a significant reduction in the expression levels of *HADHA* and *ACSL* genes (Appendix A). Specifically, HADHA plays a pivotal role in both β-oxidation and *de novo* synthesis of fatty acids as a key enzyme in fatty acid metabolic pathway. On the other hand, ACSL is involved in catalyzing lipoyl CoA synthesis, which represents an essential initial step in animal fatty acid utilization. The downregulation of transcript expression levels for these lipid metabolism-related genes may lead to a block in lipid metabolism and subsequently reduce the energy supply for normal physiological activities.

After *LvBAMBI* knockdown, transcriptome analysis also revealed a notable impact on glucose metabolism in shrimp. We observed a significant upregulation of genes associated with glucose metabolism, such as *Pck2* and *LDH*, including those involved in the tricarboxylic acid (TCA) cycle, starch, and sucrose metabolism (Appendix A). The *Pck2* gene encodes phosphoenolpyruvate carboxykinase (PEPCK), a crucial enzyme system involved in the TCA cycle and gluconeogenesis. In parallel, the *LDH* gene encodes lactate dehydrogenase, which serves as the key rate-limiting enzyme in anaerobic glycolysis and plays an important role in gluconeogenesis. The significantly upregulated expression of these genes may enhance the processes of anaerobic glycolysis and gluconeogenesis, thereby increasing glucose metabolic capacity and providing a greater energy supply to the body.

### 3.8. Effects of LvBAMBI Knockdown on Lipid Metabolism

GBA, SMPD1, and CerS are closely associated with the synthesis of ceramides. Ceramides can induce or activate certain genes (e.g*. SREBP*), thereby promoting the conversion of free fatty acid (FFA) into triglycerides for storage in lipid droplets [34]. Downregulation of these genes could significantly decrease ceramide production through both the GBA and SMPD1 pathways. Consequently, this reduction may lead to a decline in triglyceride production and potentially result in slow growth in shrimp. To further test this hypothesis, we conducted lipidomic analyses on experimental and control groups. From the lipidomic results, we observed a significant decrease in triacylglycerol (TG) content in glycerolipid metabolism after *LvBAMBI* RNAi. Additionally, there was a notable reduction in ceramide and sphingosine 1-phosphate (S1P/SPHP) levels within sphingolipid metabolism (Figure 7A,B). Triglycerides play a crucial role as energy suppliers and stores in organisms, serving as central molecules of lipid metabolism. Moreover, both S1P and ceramide function as biologically active signaling molecules that regulate various cellular processes such as cell growth, differentiation, senescence, and apoptosis. The significant downregulation of these metabolites directly disturbs nutrient accumulation and reduces energy production, resulting in impaired shrimp growth.

In the pathways of fatty acid elongation and degradation, we observed a notable reduction in hexadecanoic acid levels and overall fatty acid levels in the metabolome, respectively (Figure 7B and Figure 8A,B). These changes indicate an inhibition of both fatty acid elongation and fatty acid degradation after *LvBAMBI* knockdown. As a result, this disruption hinders fatty acid oxidation and decomposition processes, ultimately leading to a reduced capacity for lipid metabolism.

### 3.9. Effects of LvBAMBI Knockdown on Glucose Metabolism

To confirm the complementary role of glucose metabolism in meeting the energy requirement when lipid metabolism is hindered in shrimp, we conducted a metabolome analysis on both the experimental and control groups after *LvBAMBI* knockdown. We performed KEGG enrichment analysis on the identified differential metabolites and presented the top 20 differential metabolites in the pathways associated with glucose metabolism (Figure 8C,D).

In the metabolomic results, we observed a significant upregulation of the metabolites associated with glucose metabolism, including D-fructose 6-phosphate (F-6-P), pantothenol, and pantetheine (Figure 9A–C). The upregulation of F-6-P, a pivotal intermediate in the glucose metabolic pathway, directly facilitates gluconeogenesis and glycolysis. Pantothenol can be converted to pantothenic acid and then synthesized into coenzyme A, promoting the metabolism of proteins, lipids, and carbohydrates. Pantetheine is a vitamin B5 derivative that plays a crucial role in metabolism by aiding in the conversion of lipids, proteins, and carbohydrates into energy. The significant increase in the metabolic levels of all three substances indicates a substantial enhancement in glucose metabolism and increased energy production after *LvBAMBI* knockdown. However, apart from the significantly upregulated metabolites mentioned earlier, we also observed a significant down-regulation of alpha-ketoglutaric acid (AKG) (Figure 9D), a crucial intermediate metabolite in the TCA cycle. This downregulation directly inhibits the second step of the dehydrogenation reaction in the TCA cycle, resulting in decreased ATP production and subsequent inhibition of the TCA cycle. These results suggest that when lipid metabolism is blocked after *LvBAMBI* knockdown, alternative pathways such as glycolysis may be utilized to meet the energy requirements.

## 4. Discussion

### 4.1. Conservation of BAMBI Gene Structure and Function

In this study, we identified a growth-related gene, *LvBAMBI*, from the *L. vannamei* genome through SNP frequency analysis in the resequencing data and validation in individuals with different growth rates. We also conducted an extensive investigation of the gene structure, expression characteristics and function of *LvBAMBI*. The *BAMBI* genes possess a conserved Pfam_BAMBI structural domain with a signal peptide that shows high homology between invertebrates and vertebrates. Motif structure prediction suggests that the *BAMBI* genes are relatively conserved. During evolution, the motif structure of *BAMBI* genes is relatively simple in bryozoans, insects, crustaceans and echinoderms, however, it tends to become more complex in vertebrates. The presence of the conserved Pfam_BAMBI domain and transmembrane domain suggest that the *BAMBI* gene may possess common and critical functions across different species. At the same time the emergence of novel motifs in vertebrates implies their acquisition of new functionalities.

The regulatory region of *LvBAMBI* was predicted to harbor a large number of transcription factors (TFs), including CREB1, MYC, HIF1A, EZH2 and ESR1. In particular, CREB1 is closely associated with adipocyte differentiation and also plays a pivotal role in regulating cell proliferation, apoptosis, and the expression of inflammatory response factors [35]. MYC regulates cell differentiation and proliferation and is closely implicated in tumorigenesis [36]. HIF1A serves as a key transcriptional regulator of the hypoxic adaptive response associated with biological hypoxic stress [37]. EZH2, a histone methyltransferase, primarily participates in suppressing aberrant tumor cell proliferation [38]. ESR1 acts as an intranuclear estrogen receptor and controls gonadal development in organisms [39]. Analysis of these TFs suggests that *LvBAMBI* may have multiple functions, including the regulation of lipid metabolism and cell proliferation.

As a member of the TGF-β superfamily, BAMBI encodes a transmembrane glycoprotein that shares similarity with the TGF-β type I receptor. However, it lacks the intracellular serine/threonine kinase structural domain required for TGF-β signal transduction. Our analysis confirm that *LvBAMBI*′s deduced protein sequence lacks the serine/threonine kinase structural domain. Therefore, we speculate that it may play a role as a pseudoreceptor in mediating TGF-β pathway signaling. Under normal conditions, the process of TGF-β pathway signaling involves ligand binding to the TGF-β type II receptor to form a dimer, which then binds to the type I receptor for intracellular signaling. Due to the structural similarity between BAMBI and the extracellular region of the TGF-β type I receptor, BAMBI can bind competitively to the type II receptor to prevent the formation of functional ligand-receptor complexes. However, due to its lack of the intracellular serine/threonine kinase domain required for signaling, BAMBI is unable to phosphorylate downstream Smads, thereby effectively blocking the transduction of TGF-β pathway signaling.

As a transmembrane protein, BAMBI has complex functions. Its mode of action in the TGF-β and Wnt signaling pathways is similar. In the TGF-β pathway, BAMBI can either bind to the type II receptor to inhibit signal transmission or interact with another inhibitory factor, Smad7 [40], thereby collaborating to inhibit TGF-β signaling. Similarly, in the Wnt pathway, it binds to the membrane receptor frizzled (Fzd) and also interacts with the cytosolic protein dishevelled (Dvl), increasing the affinity between Wnt, Fzd, and Dvl [23]. These similar mechanisms of action do not yield identical effects on the signaling process, they hinder TGF-β signaling and promote Wnt signaling. Nonetheless, both ultimately affect cell growth.

### 4.2. The Effect of LvBAMBI on Growth

In vertebrates, the majority of studies on *BAMBI* genes have focused on mesenchymal stem cell (MSC) differentiation [41], cell proliferation and differentiation [42], adipose tissue formation [24], cellular myogenic differentiation [43], and a few researchers have focused on some of its roles in porcine granulosa cell steroidogenesis [44]. The main signaling pathways involved primarily include the TGF-β and Wnt pathway [23]. In this study, based on the expression profiles and qPCR validation results in *L. vannamei*, it was observed that the *LvBAMBI* gene exhibits widespread expression in different tissues, developmental stages, and molting stages, suggesting a complex and diverse biological function of *LvBAMBI*. Furthermore, RNAi experiments revealed that the dsBAMBI group exhibited a significantly attenuated increase in both body length and weight compared to the control group. These results provide further confirmation of previous GWAS analysis indicating that *LvBAMBI* may play a crucial role in shrimp growth.

Through transcriptome analysis after *LvBAMBI* knockdown, we identified a number of differentially expressed genes associated with muscle and growth. Notably, *kielin/chordin-like protein* (*KCP*) and *msx2-interacting protein-like* (*MINT-like*), were significantly upregulated exclusively in the dsBAMBI groups. KCP enhances bone morphogenetic protein signaling while concurrently inhibiting TGF-β signaling in cells and transgenic mice [45], MINT-like is implicated in cell development and embryonic differentiation processes, whereby elevated levels of MINT-like negatively regulate Notch signaling pathway activity thereby preventing precursor B-cells from differentiating into marginal B-zone cells [46]. The significant upregulation of these genes may inhibit shrimp growth and development, potentially accounting for the observed slow growth in the experimental group after *LvBAMBI* RNAi. Furthermore, we identified several significantly downregulated genes in the dsBAMBI group, including the muscle-related genes *actin-related protein 2/3* and *actin muscle-like*, which are structural proteins that make up muscle tissue. The transcriptional inhibition of these genes suggests a suppression of muscle protein synthesis, which could ultimately affect shrimp growth.

More significantly, through KEGG enrichment analysis, we identified several signaling pathways closely associated with growth, including MAPK pathway-fly, Hippo pathway-fly, FoxO pathway, and TGF-β pathway. The significant downregulation of genes within these growth-related pathways suggests that *LvBAMBI* RNAi inhibits the biological activities linked to these pathways to varying degrees, consequently leading to a significantly slower growth rate compared to the control group.

### 4.3. Effect of LvBAMBI on Lipid Metabolism and Glucose Metabolism

The enrichment of DEGs in biological processes (BP) and molecular functions (MF) provides a broader perspective for understanding the biological functions of *LvBAMBI*. Among the top 30 GO terms, BP and MF were mainly enriched in lipid transport, lipid localization and molecular functions related to lipid transporter activity. Knockdown of *LvBAMBI* significantly downregulated the expression levels of numerous genes involved in lipid metabolism, thereby impairing key processes such as fatty acid β-oxidation process and de novo fatty acid synthesis. As a result, the metabolism of various lipids is inhibited to varying degrees, leading to significant reductions in key metabolites essential for lipid formation, such as triglycerides. Consequently, this further reduces the energy supplied by lipid metabolism for normal physiological activities and manifests as impaired growth and development of organisms. These results indicate that *LvBAMBI* plays a crucial role in lipid synthesis, metabolism, and transport in shrimp.

In addition, prior to gonad development, *LvBAMBI* was highly expressed in the epidermis, Oka, and gill. During gonad development up to the maturation stage, this gene displayed high expressed in the ovary, testis and gill. The change in this expression pattern suggests a correlation between *LvBAMBI* and gonadal maturation. Furthermore, considering the predictions of TF analysis results, MYC may regulate *LvBAMBI* transcript levels while being associated with lipid formation and differentiation [47,48]. It is hypothesized that there is an elevated level of lipid metabolism during the gonadal developmental stage, which may be closely related to the lipogenic effects of BAMBI.

KEGG pathway enrichment analysis also revealed that the upregulated DEGs were involved in several glucose metabolism-related pathways, including glycolysis/gluconeogenesis, glycosaminoglycan degradation, glycosaminoglycan biosynthesis, and the TCA cycle. This suggests that knockdown of *LvBAMBI* increased glucose metabolism in shrimp, potentially compensating for the energy deficit resulting from impaired lipid metabolism. Glycome analysis demonstrated a significant up-regulation of F-6-P expression which is involved in the regulation of glycolysis and gluconeogenesis. These findings support our previous hypothesis regarding the potential enhancement of certain glucose metabolic processes to compensate for the energy lost after *LvBAMBI* knockdown.

We also found some changes in pathway genes associated with protein processing and transport in the KEGG pathway enriched by down-regulated DEGs, including protein processing in the endoplasmic reticulum, protein export, ABC transporters, etc. This compelling evidence suggests that *LvBAMBI* may also serve as a mediator in the regulation of protein metabolism.

## 5. Conclusions

In this study, we conducted a comprehensive investigation of the *BAMBI* gene in *L. vannamei* using genomic, transcriptomic, metabolomic, and RNAi experiments. Knockdown of *LvBAMBI* primarily affected growth by modulating lipid metabolism, glucose metabolism, protein transport, and immune-related biological processes. This study provides a valuable basis for further research and utilization of *BAMBI* genes in shrimp and crustaceans.

## Figures and Tables

**Figure 1 animals-14-01074-f001:**
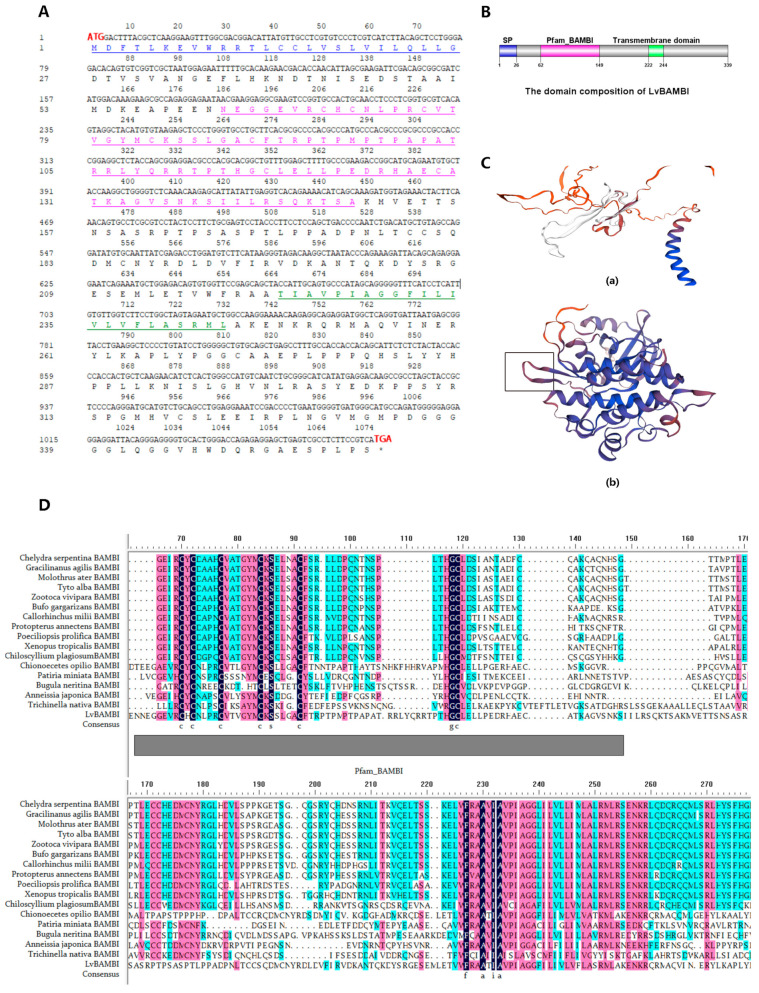
Structural characteristics of *LvBAMBI*. (**A**) Nucleotide and amino acid sequences of *LvBAMBI*. The start codon and stop codon are shown in red. The predicted signal peptides are shown in green. Pfam_BAMBI is labeled in pink. The transmembrane_domain was highlighted with underlined and labeled green. (**B**) The domain composition of LvBAMBI. SP, signal peptide. (**C**) 3D structure prediction of LvBAMBI protein. (**a**) The predicted protein three-dimensional (3D) conformation of LvBAMBI. (**b**) The predicted 3D conformation of TGF-β type I receptor. The white area in (**a**) had a similar structural pattern to the rectangular area in (**b**). (**D**) Multiple sequence alignment analysis of BAMBIs. The highly conserved sites were depicted in black with corresponding amino acid names provided at the bottom, and partial sequences exhibiting significant homology were highlighted in pink and blue.

**Figure 2 animals-14-01074-f002:**
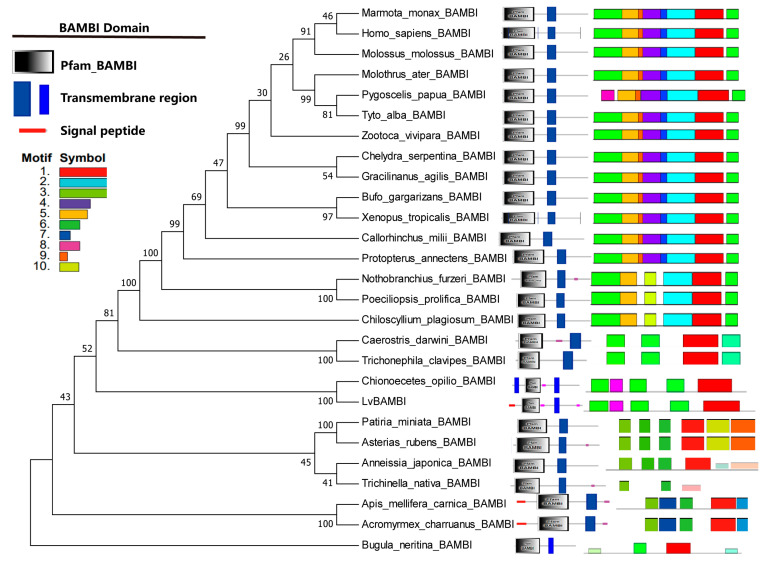
Phylogenetic tree of BAMBIs and their gene structures. The BAMBIs domains from the SMART program are on the right of the phylogenetic tree. Bootstrap values are given at each branch node. The pink lines in *C. darwini*_BAMBI, *C. opili_*BAMBI, LvBAMBI, etc., represent low complexity regions.

**Figure 3 animals-14-01074-f003:**
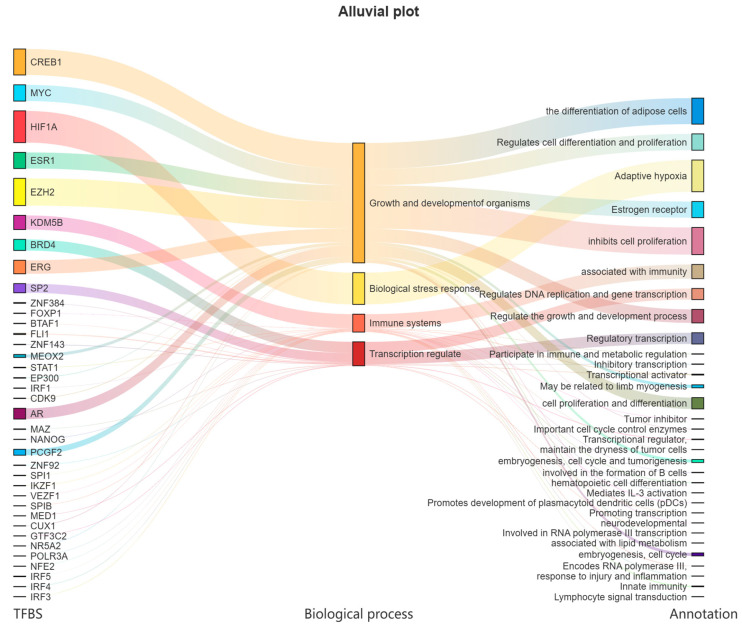
Transcription factors predictive analysis of *LvBAMBI* and their potential functions. The transcription factor prediction results are sourced from Ensembl and hTFtarget databases.

**Figure 4 animals-14-01074-f004:**
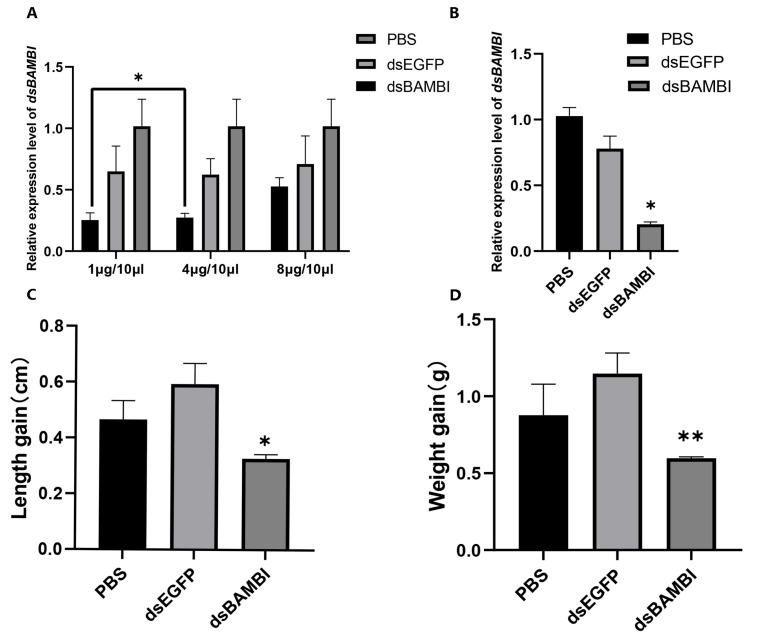
Efficiency detection and growth characteristics of *L. vannamei* after *LvBAMBI* RNAi. (**A**) Knockdown efficiency of *LvBAMBI* in shrimp at 48 h after injection of different doses of dsRNA. (**B**) Relative expression levels of *LvBAMBI* after 16 d of RNAi at the optimal interference dose. (**C**) Changes in shrimp body length after 16 d of continuous *LvBAMBI* RNAi. (**D**) Sustained changes in body weight after 16 d of *LvBAMBI* RNAi. The expression of target genes was detected by qRT-PCR and normalized to the 18S rRNA gene as the internal reference. These results were based on three independent biological replications and are shown as mean values ±SD. Significant differences in the gene expression levels between the three treatments are shown as * *p* < 0.05 and ** *p* < 0.01.

**Figure 5 animals-14-01074-f005:**
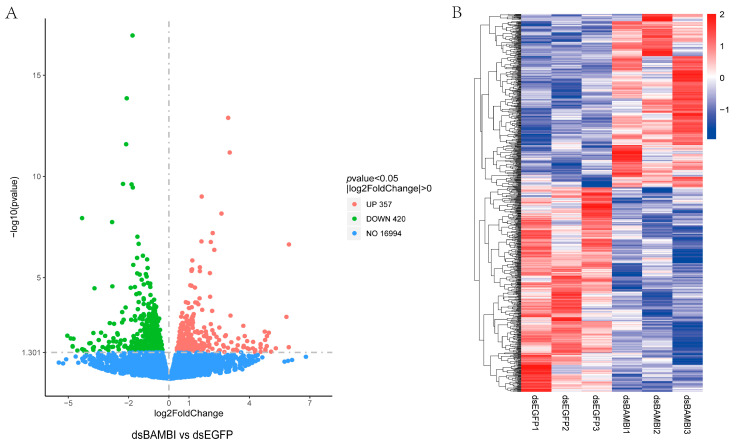
DEGs of hepatopancreas in *L. vannamei* after *LvBAMBI* RNAi. (**A**) The number of DEGs among hepatopancreas. (**B**) The heat map of DEGs among three parallel hepatopancreas samples. Abscissa for |log2FoldChange| > 0, Ordinate for *p*-value < 0.05.

**Figure 6 animals-14-01074-f006:**
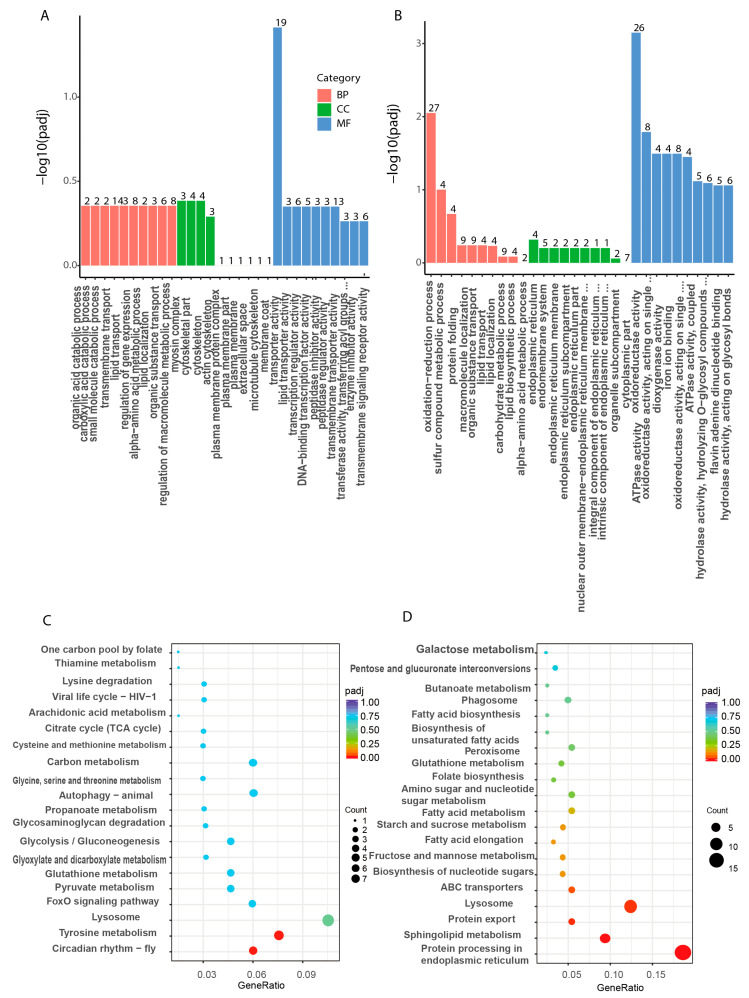
Top GO terms of DEGs and the pathways of KEGG enrichment analysis in the hepatopancreas of *L. vannamei* after *LvBAMBI* RNAi. (**A**) The top 30 GO terms of biological functions of upregulated. (**B**) The top 30 GO terms of biological functions of downregulated. The horizontal axis represents the name of the pathway and the vertical axis shows the gene ratio. (**C**) KEGG enrichment pathway analysis of upregulation genes. (**D**) KEGG enrichment pathway analysis of downregulation genes. (**C**,**D**) The size of the point denotes the number of DEGs while the color corresponds to the Q-value. A deeper color represents a smaller Q-value and indicates more significant pathway enrichment.

**Figure 7 animals-14-01074-f007:**
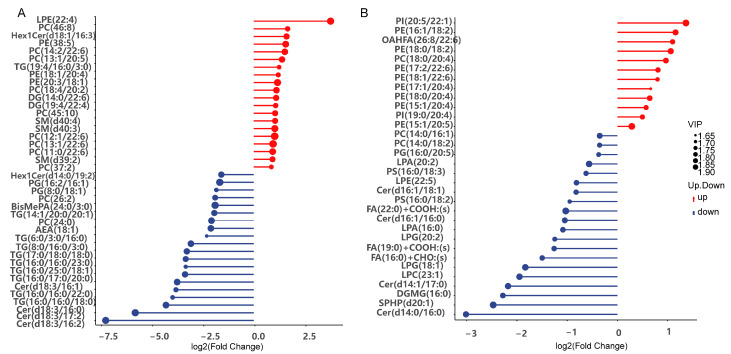
The top 20 lipid compounds for up- and downregulation for the matchstick plot presentation. (**A**) Trends of up- and downregulation of TG and Cer in the differential lipid metabolites of experimental and control groups and changes in the multiplicity of the difference. (**B**) Trends of up- and downregulation of Cer, FA and SPHP in the differential lipid metabolites of the experimental and control groups and changes in the multiplicity of the differences. The color of the dots represents up- and downregulation, blue represents downregulation, red represents upregulation; the length of the rods represents the magnitude of log2 (Fold Change); the size of the dots represents the magnitude of the VIP value.

**Figure 8 animals-14-01074-f008:**
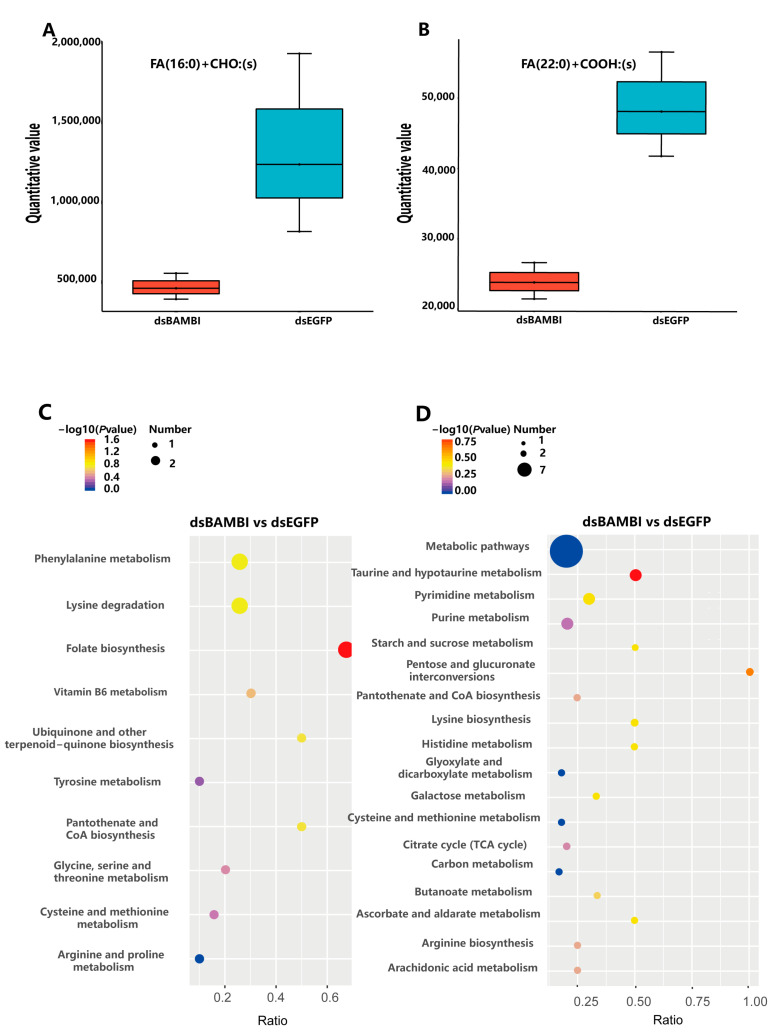
A comparative analysis of lipid metabolites in the experimental and control groups was performed using box plots and KEGG enrichment analysis. (**A**,**B**) Boxplot of lipid metabolite differences between different groups. Vertical coordinates are quantitative values of metabolites and horizontal coordinates are different groups. (**C**,**D**) KEGG enrichment bubble plots of the top 20 differential metabolites. The vertical axis represents the name of the pathway and the horizontal axis shows the gene ratio. The size of the plot denotes the number of DMs while the color corresponds to the Q-value or *p*-value. A deeper color represents a smaller Q-value or *p*-value and indicates a more significant pathway enrichment.

**Figure 9 animals-14-01074-f009:**
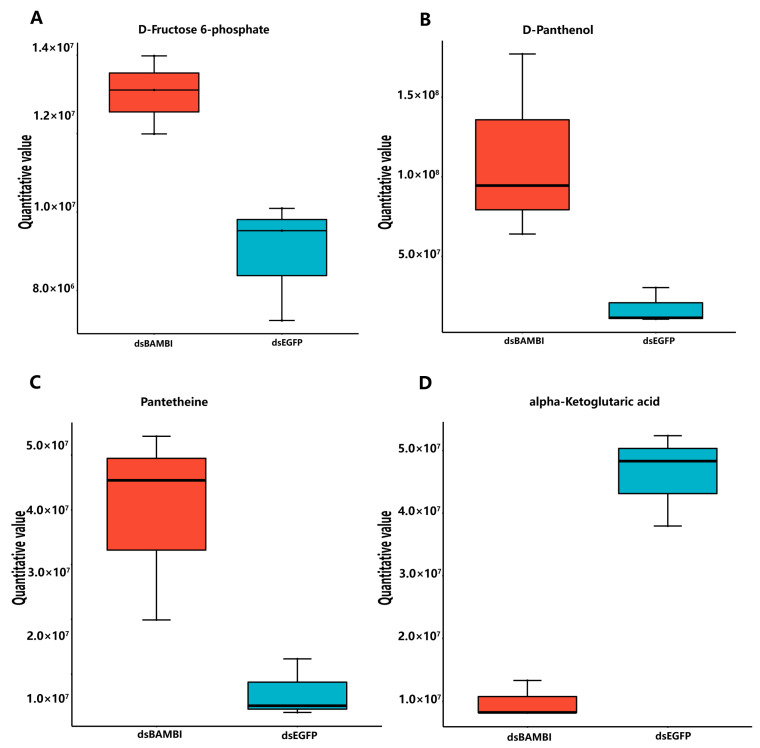
Boxplot of differential metabolites between different groups. (**A**) D-Fructose 6-phosphate boxplot (**B**) Panthenol boxplot (**C**) Pantetheine boxplot (**D**) alpha-Ketoglutaric acid boxplot. The vertical coordinate is the quantitative value of the metabolite and the horizontal coordinate is the different groupings.

**Table 1 animals-14-01074-t001:** Some DEGs in hepatopancreas after *LvBAMBI* RNAi.

Gene	Annotation	Log2(foldchange)	Regulation	Possible Functions
LOC113810077	pancreatic lipase-related protein 2-like	2.055644473	up	lipid metabolism
LOC113805268	group XV phospholipase A2-like	1.512752809	up	
LOC113816418	elongation of very long chain fatty acids protein 6-like	−1.109271264	down	
LOC113807853	17-beta-hydroxysteroid dehydrogenase 13-like	−1.278968958	down	
LOC113815038	sphingomyelin phosphodiesterase-like	−1.520403624	down	
LOC113814209	TLC domain-containing protein 2-like	−1.529060798	down	
LOC113808685	CYP450	−1.642795706	down	
LOC113815027	apolipoprotein D-like	−1.833408201	down	
LOC113813817	apolipoprotein D-like	−2.306012563	down	
LOC113825958	Leukocyte Elastase Inhibitor	2.011631663	up	immune-related genes
LOC113823643	hemocyanin C chain-like	1.497524117	up	
LOC113823622	hemocyanin C chain-like	1.108584479	up	
LOC113825585	cathepsin L1-like	1.099745679	up	
LOC113823633	hemocyanin C chain-like	1.036162219	up	
LOC113807239	Beta-1,3-glucan-binding protein	−1.589632096	down	
LOC113807240	Beta-1,3-glucan-binding protein	−1.606268937	down	
LOC113828583	metabotropic glutamate receptor 3-like	5.937174763	up	intracellular signal transduction, and transportation
LOC113805618	sorting nexin-18-like	2.99536541	up	
LOC113811177	SLC19A1	2.14994886	up	
LOC113807061	sialin-like	1.603719878	up	
LOC113800776	neurexin-1-like	1.363895445	up	
LOC113810616	hemocyanin C chain-like	1.096986803	up	
LOC113805473	YvkC-like	−1.052007081	down	
LOC113821593	phosphoenolpyruvate synthase	−1.071866976	down	sugar synthesis and breakdown
LOC113806982	delta-1-pyrroline-5-carboxylate synthase-like	−1.79551763	down	
LOC113814709	Hydroxyacid-oxoacid transhydrogenase	−2.835674612	down	
LOC113828937	period circadian protein-like	1.590523236	up	Other metabolism related genes
LOC113825869	trichohyalin-like	1.283633328	up	
LOC113812271	methionine synthase-like	−1.283853613	down	
LOC113822192	SMCT1-like	−1.478478109	down	
LOC113805739	chymotrypsin BII-like	−1.820489948	down	protein metabolism
LOC113822498	ubiquitin-40S ribosomal protein S27a-like	−3.724189533	down	
LOC113814568	kielin/chordin-like protein	1.127120613	up	Genes related to the growth and development of organisms
LOC113821325	leucine-rich repeat transmembrane neuronal protein 4-like	−1.036589648	down	
LOC113820707	protein patched homolog 2-like	−1.599886556	down	
LOC113826172	HSP70	−1.132309417	down	Genes involved in cellular homeostasis
LOC113806042	HSP83/HSP90	−1.529544744	down	
LOC113813873	heme-binding protein 2-like	−1.925589424	down	
LOC113828559	branchpoint-bridging protein-like	−1.31764033	down	Transcription regulation of related genes

## Data Availability

The datasets presented in this study can be found in online repositories. The names of the repository/repositories and accession number(s) can be found in the article/Appendix A.

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
