# Peer review of "Identification of Growth-Related Gene BAMBI and Analysis of Gene Structure and Function in the Pacific White Shrimp Litopenaeus vannamei"

_animals, 2024, doi:10.3390/ani14071074_

Round 1

Reviewer 1 Report

Comments and Suggestions for Authors

General comments

In this study, SNP variation analysis was performed and subsequently a growth-related gene BAMBI (LvBAMBI) was identified from the Pacific white shrimp Litopenaeus vannamei. Gene silencing of LvBAMBI by RNA interference inhibited shrimp growth significantly. Moreover, RNAseq analyses using the gene knockdown shrimps suggested that LvBAMBI regulates lipid metabolism, glucose metabolism, and protein transport. The methods in this study are technically correct and the data for the exhaustive analyses is sufficient. Therefore, the article might be suitable for the publication in Animals.

Specific comments

1) Lines 75-83, the explanation of BAMBI is insufficient. Adding details such as BAMBI's biological functions, expression tissues in vertebrates, and the phenotype when knocked out would help the reader understand BAMBI better.

2) Lines 438-447, the reason for determining the optimal dose of 4 μg for dsBAMBI are not mentioned in this section, so an explanation needs to be added. The injection of dsBAMBI resulted in a decrease in the body length and weight of the shrimps. However, it is not mentioned whether there were any changes in the amount of food consumption or the frequency of molting. It would be better to include this information.

Author Response

Thank for your comments concerning our manuscript entitled “Identification of Growth-related Gene BAMBI and Analysis of Gene Structure and Function in the Pacific White Shrimp Litopenaeus vannamei” (Manuscript ID: animals-2892219). These comments are very helpful to improve the quality of the manuscript. We have read all comments carefully and have made conscientious correction, further clarify the logic of writing for improving the quality of the manuscript. Here we response the comments with point by point.

Reviewer 2 Report

Comments and Suggestions for Authors

In general, this manuscript titled as” Identification of Growth-related Gene BAMBI and Analysis of Gene Structure and Function in the Pacific White Shrimp Litopenaeus vannamei” reports the results of a series of experiment presenting data on a potential growth-related gene in the mostly cultured species Litopenaeus vannamei. Due to the scarcity of such information, the presented data will provide important information on the underlying molecular mechanism of shrimp growth for designing breeding program in shrimp aquaculture. Nonetheless, several important revisions need to be made before this manuscript can be published.

So many abbreviations were used which need to be presented in the elaboration form first, such as Myostain (Mstn) in line 66 and transforming growth factor (TGF) in line 71.

Lines 60-83: It is very difficult to stay focused on what the authors wanted to say and why they targeted specifically the "BAMBI" in this study? Where they have discussed so many growth-related pathways or genes throughout these two paragraphs.

Line 58: “several relevant studies have already been reported [16,17]” explain a bit more in what aspect that are relevant.

Line 86: “Subsequently, we cloned the gene”, did they clone?

Line 97: “Experimental shrimps were cultured…... “. Need a brief description of them, male or female or all females? Sub-adult or adults (body weight and age in days or months) and sources, like from a commercial hatchery or from another shrimp research institute, etc. In fact, it is also unclear here why the animals were reared for? Only for RNAi in section 2.6, right?

Line 114: What did they mean by “resequencing data”? It needs explanation.

Line 132-140: the explanation of selecting the BAMBI gene from the screened 25 genes is not easily understood.

Line 222, in 2.7. Real-Time Quantitative PCR: Mention the length of sequence to be amplified for both genes. It was mentioned about only hepatopancreas tissue RNA was isolated and the relative expression of LvBAMBI was detected by SYBR Green-based quantitative real-time PCR (qPCR) for experimental and control groups after RNAi treatment. Perhaps for determining changes in the metabolism pathways, the expression pattern of hp-tissue was only tested after RNAi treatment. However, in Fig. S1B, in adult, the LvBAMBI was highly expressed in “In, Oka, and Gi tissues”. It would be interesting to compare with that after RNAi-treatment. What do the authors think about it?

 Line 375, 3.3. section “Phylogenetic Analysis of BAMBI Genes”: Mention how many bootstraps were used? It is better to give the sequence accession number of all used BAMBI sequences in the Phylogenetic analysis in a supplemental table.

Line 378: Correct Fig.2 to Fig. 2.

Line 380: Use arrows or colored branches to indicate nodes so that readers can easily understand them.

Line 383-388: Mention with color into a parenthesis like, an N-terminal signal peptide (SP(Sec/SPI)) (red line), a Pfam_BAMBI (gray box), BAMBI domain (gray line), and a C-terminal transmembrane structural domain (blue box) (Fig. 1B). Now, what is the pink line here in C. darwini_BAMBI, C. opilio_BAMBI, LvBAMBI, etc? Figure 2 needs more explanation to understand what the authors wanted to present here.

Line 418, section “3.5. Expression Pattern of LvBAMBI”: In Fig. S1A-C, the Y-axis is labelled as FPKM only. Doesn’t it Log2FPKM?

Line 430-432: In S1D, how did they normalize the relative expression level of LvBAMBI in 12 adult tissues of L. vanname? If it’s with the 18S expression, mention it on the graph like, “relative expression to 18S or so on”.

Fig. S1: In S1B, the transcriptional profile of LvBAMBI was examined in 16 different tissues, but in the S1D, the results of qPCR of 12 tissues were presented. Explain what the criteria were for giving emphasis on tissues selection. Moreover, was the Fig. S1D results of the RNAi treatments in the section “2.7. Real-Time Quantitative PCR”? Provide a clear explanation for easy understanding.

Line 437: It was discussed that the gonadal development within their sample groups were at different developmental stages. Again, there is no data on the condition of sampled shrimps’ selection, like, male or female (all). Didn’t they initially select shrimp with apparently the same condition? Or did they select randomly? It needs more details on choosing the experimental animals. 

Comments on the Quality of English Language

Good.

Author Response

Thank for your comments concerning our manuscript entitled “Identification of Growth-related Gene BAMBI and Analysis of Gene Structure and Function in the Pacific White Shrimp Litopenaeus vannamei” (Manuscript ID:animals-2892219). These comments are very helpful to improve the quality of the manuscript. We have read all comments carefully and have made conscientious correction, further clarify the logic of writing for improving the quality of the manuscript. Here we response the comments with point by point.

Reviewer 3 Report

Comments and Suggestions for Authors

This manuscript is about the molecular identification of Growth-related Gene BAMBI from white shrimp L. vannamei and its characterization by bioinformatical and RNAi tools. They followed traditional genomic tools to characterize BAMBI gene and RNAi experiment showed significant decrease in both body weight and total body length. The overall experimental procedure and analyses were acceptable and the results were sound, which indicated that LvBAMBI be involved in growth of L. vannamei shrimp. This result would helpful for those who study the growth of decapod species.

However, there are several issues to be resolved.

1. In figure 4, I could not find any information what kind of organs were used and why they used it. In order to know the effects of RNAi two or three the  organs should be tested in which LvBAMBI is highly expressed. Besides days post injection also impact on the interference . Please describe them.

2. Figure 5, I have no idea why author used this tissue to see transcriptome. There are more other organs in which LvBAMBI expression is high. please describe them in the text.

3. Figure 4C & D I also have no idea why they tested only 19 days for RNAi experiment. Considering molting cycle it would be clear for longer time of experiment. please explain about this issue. 

Author Response

(The authors gave the same response as above.)

Round 2

Reviewer 3 Report

Comments and Suggestions for Authors

Authors answered all my questions and now I believe acceptable for publication as it is 

Author Response

Thank you for your suggestions and comments, which are very helpful in improving the quality of this manuscript and deepening our understanding of this study. Thank you again.